# Medicinal Plants as a Natural Greener Biocontrol Approach to “The Grain Destructor” Maize Weevil (*Sitophilus zeamais*) Motschulsky

**DOI:** 10.3390/plants12132505

**Published:** 2023-06-30

**Authors:** Ompelege Jacqueline Phokwe, Madira Coutlyne Manganyi

**Affiliations:** Department of Biological and Environmental Sciences, Faculty of Natural Sciences, Walter Sisulu University, Mthatha 5117, South Africa; ophokwe@wsu.ac.za

**Keywords:** maize weevil, medicinal plants, *Sitophilus zeamais*, biopesticides

## Abstract

According to the United Nations (UN), the global population may skyrocket to 9.8 billion people in 2050 and 11.2 billion in 2100, placing an overwhelming burden on food security as the world will have to meet this growing demand. Maize is the largest staple grain crop produced in developing countries. The maize weevil, *Sitophilus zeamais*, is one of the most destructive post-harvest pests of stored cereals and grains. The maize weevil contributes up to 40% of total food-grain losses during storage, mainly in developing countries. Current synthetic pesticides are ineffective, and, moreover, they raise serious environmental safety concerns as well as consumer health hazards. Drawing from past oversights and current environmental realities and projections, the global population has been switching to green living by developing sustainable strategies. In our context, these new greener strategies include the utilization of medicinal plants to control maize weevil infestation, which unlocks unlimited innovative possibilities, and, thus, improves the yield, quality, and safety of maize. Medicinal plants are less toxic, easily biodegradable, and capable of protecting grain from pests. This paper systematically outlines the literature on host plants as well as the feeding and associated diseases of the maize weevil. In light of this, we cement medicinal plants as excellent candidates in the pursuit of greener, sustainable, more potent, and cost-effective pesticides.

## 1. Introduction

The current food security crisis has led to vigorous debate on how best to feed the growing global population. It has been reported that the maize weevil contributes up to 40% of total production in food–grain losses during storage, mainly in developing countries. The *Sitophilus zeamais* Motschulsky (Coleoptera: Curculionidae) maize weevil is one of the most significant pests of stored grains and is mostly accountable for maize damage [1]. In 48 days, 18.3% of losses are predicted to result from an average of two insects per grain [2]. In addition, the significant grain damage may lead to the reduction in nutritional quality, weight, and germination rates of seeds, and it may also affect human health. Insects can also transmit pathogenic fungi, including *Aspergillus flavus*, which has a link to several types of bacteria. Larvae and adults are responsible for most of the damage to the grains, and 50% of the eggs can be laid within the first 5 weeks of an adult’s life [3]. The female drills the grain in order to create tiny, chewed chambers for oviposition, which are then sealed by a secretion, protecting the creature so that it can continue its life cycle there. Consequently, the majority of solutions emphasize adult control [2]. Ecological and occurrence data have shown that maize weevil thrives in regions of the world that are warm and tropical, particularly in those where maize is grown [4]. Moreover, the maize weevil is shipped in grain shipments to every country in the world, and it can infect food where grain moisture and temperature are favourable. There is also a growing concern regarding the development of resistance in numerous stored product insects, including the maize weevil, in grain stores and flour mills as well as in the food industry [5,6]. In South Africa, maize is used in livestock feed, and a significant portion of the country’s households depend on it. The United States is the biggest producer of maize in the world with 2000 million tons per annum, followed by China with 717 million tons per annum [7]. In Africa, South Africa is the biggest producer of maize with an annual production of approximately 10 million tons, although it can vary depending on the rainfall. Furthermore, maize is used industrially to manufacture an array of products, and it is traded both locally and internationally for financial gain and economic growth [8].

Severe food insecurity at the global level has been rising in recent years, mainly in Africa and Latin America [9]. However, as these grains are collected, stored, processed, and marketed to customers, 10–30% of these significant cereal crops, such as maize, are lost to insect pests after harvest. Although this process can be effectively controlled by synthetic chemical pesticides [10], the majority of farmers in Africa are resource poor and have neither the means nor the skills to obtain and handle pesticides appropriately [5]. The prohibitive costs of commercial synthetics, the increasing development of insect resistance to pesticides, toxicity concerns, and an often erratic supply of pesticides have given impetus to the search for alternative insect control measures [11]. About 80% of pesticides that are used lead to water contamination, thus affecting animals, farmers, and consumers of agricultural products, and putting them at risk of serious health problems. A body of evidence has shown that farmers exposed to inorganic pesticides by spraying crops on a regular basis have developed many health problems, some of which are fatal [12,13]. 

In view of the above, it is essential to develop more eco-friendly, greener, and sustainable alternative pest management strategies. Medicinal plants have been the focal point of eco-friendly and sustainable research investigations. Despite this, research on medicinal plants as a pest control substitute is in short supply, and so is knowledge about this possibility. Therefore, we examine the utilization of medicinal plants as a natural and greener control approach compared to harmful synthetic pesticide. In order to bridge the research gaps, the current review investigates the various host plants, feeding, and disease transmission of the maize weevil. After laying this groundwork, the current status of pest management and resistance is discussed as a matter of concern. Ultimately, we provide sound evidence to support the use of medicinal plants as biocontrol agents against the maize weevil in order to contribute to sustainable greener living.

## 2. Methodology

During this study, we conducted an in-depth review of important published papers about maize weevil and the utilization of medicinal plant as biopesticides as promising control strategy. We screened abstracts and chose articles before reading full publications in order to streamline the search. Reputable search engines such as Scopus, Google Scholar, ScienceDirect, PubMed, and Web of Science were utilized to search, examine, and select relevant research papers.

## 3. Host Plants and Feeding

Based on the knowledge [14,15,16], the maize weevil is well known to consume and infect a wide range of hosts including maize, wheat, rice, sorghum [17,18], oats, rye, cottonseed, buckwheat, peas, and barley [19]. Although it has been observed that maize weevils prefer whole grains, they also feed on pet food, pasta, other processed grains, and even fruits [20,21]. In view of this [22], we concluded that maize weevil pests infect biscuit crops and non-cereals crops, such as yam and cassava chips as well as other host plants. Both adults and larvae are capable of surviving on a variety of grain-based foods. Furthermore, it was reported in North America that maize weevils feed on stored grain and on fruits such as peach or apple, and that they inhabit forests or grasslands in southern Japan, where they feed on flowers in the spring [23]. In general, the maize weevil’s success depends on its capacity to find and recognize an appropriate host species, access s plant’s fitness, and efficiently use the resources of the plant [23]. Food materials that contain more than 10% water are more likely to favour Sitophilus species population dynamics [24].

By reducing their components, the maize weevil has an impact on processed food materials and the nutritional content of the food materials [25]. This makes sense, since the maize weevil is found in warm, humid regions all over the world, particularly where maize is grown [26]. Grain quantity has an impact on the biology of *Sitophilus zeamais*, including oviposition, distribution of eggs, adult emergence, body weight, and sex ratio. In general, the maize weevil can infest grains prior to harvest [27,28]. Crop damage occurs in various stages in the life cycle of the maize weevil. When the female weevil penetrates, a single egg is laid inside the grain’s kernel, resulting in damage to the grain [29]. The egg is then shielded by a waxy secretion that seals the hole and hardens. When the larva hatches, it eats the grain’s pulp. The embryonic stages grow inside the grain, protecting them from pesticide contact, hence the fact that chemical treatment of this pest is only effective against the adult weevil [30].

Meanwhile, the maize weevil is causing serious economic losses as well as crop weight loss, quality loss, and disease transmission through fungal growth. It can even destroy grains that have been stored in all types of storage facilities by increasing the amount of free fatty acids [31]. The establishment of secondary and mite pests and pathogens may be aided by the invasion of this primary colonizer. As an invader, the maize weevil enters packages through openings that have already been made due to subpar seals, openings made by other insects, or mechanical damage [32]. Maize is a global industrial raw material and is also utilized in various manufacturing products such as producing alcohol (breweries) [33,34,35]. The main tool in the battle against weevil infestation in maize is currently conventional synthetic pesticide [1]. In most cases, pesticides are applied to the grain when it is placed into silos or as a surface treatment after storage in order to control the problem [36]. However, a body of evidence has shown that the current pesticides are ineffective against resistant pests. In addition, there is a great concern regarding the negative impact to the environment with respect to hazards, and the science community is similarly unsettled about the severe side effects on non-target organisms as well as the extravagant price tag [37]. Several reports have shown that the maize weevil affected maize in Southern and Eastern Africa, Central America, and Mexico [17,38]. In addition, the maize weevil has been found in a variety of crops, such as wheat in India, Australia, most of Europe, Northern Asia, and Northern Africa [39], as well as rice in China [40]. The loss of sorghum and oats were reported, too, in South Asia and Sub-Saharan Africa [41] and in Central Europe, respectively [42]. Europe described having barley losses, as well, due to the maize weevil [43]. The maize weevil thus affects a broad spectrum of host crops across the world.

## 4. Disease Transmitted

After the maize weevil infects the crops, it raises the grain’s temperature and moisture content, which provides suitable conditions for fungal contamination, thus creating an issue of fungal toxins. They are transmitted to humans mainly by consumption of maize. Simultaneously, dry maize is not suitable for human consumption, but it is exported for animal feed [44]. Based on research, maize weevils are reservoirs for a great microbial diversity mass of pathogenic fungi, such as a *A. niger*, *A. candidus*, *Penicillium islandicum*, *Paecilomyces*, *P. citrinum*, *Acremonium*, *F. semitectum*, *Epicoccum*, yeasts, and many bacteria [45]. This species thrives on leftover foods (such as rice) as well as on a variety of agricultural products (such as peanuts, dried corn, millet, tree nuts, and cotton seeds). *A. flavus* dispersal and subsequent aflatoxin contamination have been linked to a variety of insect species. The spread of the green fungus in stored maize seed is a significant issue, and research into the metabolic processes of weevils linked to the spread of the fungus’ spores is crucial [46,47], especially as it has been found that the maize weevil can contribute significantly to increased Aspergillus flavus infection on corn ears by transporting spores and damaging corn kernels. Another study has reported that grains heavily infected with storage fungi had higher than average insect populations [48], and then insects may make corn more likely to become infected by storage fungi by causing damage to the grain or else by making it more susceptible to fungal attack [49]. Storage conditions are typically warm and humid, which provides an ideal environment for fungal growth, ensuing that mycotoxins are produced. Barney et al. [50] reported that the colonization of *Aspergillus flavus* and *Aspergillus parasiticus* usually starts in the maize ears before spreading. Mycotoxin contamination, then, is considered a massive threat to both human and animal health as well as food security.

## 5. Current Pest Management and Its Challenges

In the past two decades, a few strategies have been implemented, including implemented integrated pest management (IPM), in order to control the maize weevil as well as other pests, and have caused the least amount of environmental harm. Furthermore, the scientific community incorporates compatible preventive and curative pest control as part of the IPM as an affordable approach [51]. On the other hand, environmental factors including temperature, relative humidity, photoperiod, food quality, and quantity all play a significant role in insect development. The physiological functions of the insects are also impacted by these variables [52]. In particular, the maize weevil insect’s reproductive and feeding activities have reduced maize production by causing grain loss in storage, which can reach 90% in unprotected grains [1].

In developing nations, transportation, poor storage structures, and prolonged storage promotes favourable conditions for the maize weevil to strive [10]. In this part of the world, many smallholder maize farmers reside in inaccessible, remote areas with inadequate road systems, making it challenging for them to get their produce to market. Because of this, farmers must store their grain for a considerable amount of time, often 6 to 8 months after harvest, and the maize weevil poses a threat during this time [6]. To lessen weevil infestation in storage, it is essential to maintain excellent hygienic practices. Before storing new products, fully cleaning the warehouse and the containers is often forgotten, and, as a result, it has been claimed that the leftovers from infected products can cause fresh infestations [53]. As a result, according to [14], adopting good hygiene procedures in homes and by farmers and traders is advised, as this will significantly lessen the likelihood of new maize weevil infection caused by leftovers. It has been emphasized, too, that grain conditioning prior to storage is very important [14]. Kernel hardness, a foundation of resistance even among resistant maize genotypes or variations, is constrained by moisture content. Moisture levels exceeding 16% make some varieties of corn vulnerable to attack by storage insects such as the maize weevil [54], and this creates the need for grain conditioning before storage [55].

According to Nwosu [14], it is preferable to measure the grain batch’s moisture content before the experiment and the grain batch’s moisture content after the experiment, and to then account for moisture gain or loss. Crop handling after harvest is typically inexpensive and practicable using traditional methods. To lessen insect activity during storage, various conventional materials are frequently added to the grain [56]. Furthermore, there are traditional ways of protecting stored grains, such as silos. A metal silo is a cylindrical structure constructed from a galvanized iron sheet and hermetically sealed. Metal silo technology has proven to be effective for protecting the harvested grains from attack, not only from the storage insects but also from rodent pests [57]. A metal silo is airtight, and it therefore eliminates oxygen inside, killing any insect pest that may be inside [3]. Overall, then, many methods, which include chemical, physical, and biological ones, have been used to drastically reduce the infestation of this pest [58], but the modern agricultural system has still used pesticides as the most reliable control method for pests.

Starting in the 1960s, the use of pesticides in farming operations came under heavy scrutiny when their unintended effects on the ecosystem became widely recognized [59,60,61,62,63,64]. The favoured tool in the battle against weevil infestation in stored maize is a conventional synthetic pesticide [1] as a chemical control strategy. Even though pesticides have been effective at reducing pest populations, they have detrimental effects on the environment, human health, animals, and plants. Since the 1950s, ancient storage techniques and a wide variety of insecticides (often in the form of fumigants) have been used to protect maize grain from pests [65,66], but a body of evidence has shown that the usage of pesticides is accompanied by a number of issues, including, among others, the emergence of pests that are resistant, the eradication of natural enemies, the extinction of non-target species, and the contamination of both the environment and the food [67,68,69]. Some pesticides have negative effects on the nervous, renal, respiratory, and reproductive systems of men and women according to [70], which is a product of the fact that there are basic similarities between the nervous systems of the mammalian and the insect. Indeed, pesticides (organochlorides, Ops, and carbamates) that are designed to attack an insect’s nervous system are capable of producing acute, chronic neurotoxic effects in mammals [71,72]. In addition to having a direct negative effect of active substances on both human and animal health, the excessive or improper use of synthetic pesticides is also associated with the following effects:❖Pesticide resistance in some pests.❖Water, soil, and air contamination that transfers chemical residues along the food chain.❖Reduction of biodiversity and nitrogen fixation.❖Destruction of marine and bird life and/or a contributing factor in the genetic defects in subsequent generations.❖Non-target organisms are affected even though they might be beneficial to the crops. 

In one study [73], Zeolite Slovenia (which contains 63.37% SiO_2_) and Zeolite Serbia (which contains 60.16% SiO_2_), two natural zeolites that were utilized to suppress maize weevil, outperformed a synthetic zeolite, Asorbio. The treatment using the synthetic zeolite Asorbio^®^, which was supplemented with SiO_2_ and Al_2_O_3_, was shown to have the lowest insecticidal activity. In another study [74], which was carried out in order to evaluate the impact of zeolites on the mortality of the maize weevil (*Sitophilus zeamais* Motschulsky) adults. The efficacy of natural zeolites (Slovenian and Serbian) and synthetic zeolites (‘Asorbio’) as well as Diatomaceous Earth (product SilicoSec^®^) were tested. The use of natural zeolites proved to be efficient as stored product protectant, the comparison of natural zeolites, treatments with “Asorbio” resulted in the lowest mortality of maize weevils. Synthetic insecticides could significantly reduce the amount of storage losses brought on by insect pests [74], then, although we should note the drawbacks of using synthetic pesticides, including toxic residues, worker safety, and the increased costs of application [75]. Drawing from several studies, pesticide residues have been detected in groundwater, downstream from livestock and crops [76,77,78]. In addition, non-selective pesticides kill beneficial insects and non-targeting organisms, thereby causing an imbalance in the ecosystem [79]. This has prompted more research on eco-friendly, cheaper, and safer plant products that include extracts, oils, and powders [61,80].

The use of biological control provides a suitable and greener approach to control pests. Biological control is a long-term pest control method that involves introducing an exogenous biological agent into an environment with the intention of permanently establishing it there. Additionally, biological control is a highly effective pest management strategy, especially when compared to chemical pesticides. For example, this strategy includes several management techniques that safeguard human health and the environment. It has been suggested that biological control improves crop fitness, and ultimately has a negative impact on pests [81]. In a previous study, some insect eggs, such as the eggs of the maize weevil, could survive for a week or more in a −20 freezer inside de-infested containers. Furthermore, these eggs were invisible to the naked eyes [82]. However, biological control techniques are currently used much less frequently than pesticides, despite being a less expensive and environmentally harmful technology. This is mostly because biological management is more detailed and requires more time to produce the desired results and finances [83]. According to Mascarin et al. [84], several investments have been made by some agricultural companies to produce fungicides that are currently being used in both organic and conventional plants. Subsequently, the call for a more sustainable integrated pest management program has been answered. To contribute to this development, several advantages of using biological control are listed below [85,86]:

The findings of Del Arco et al. [87] showed that the parasitoid *Anisopteromalus calandrae* (Hymenoptera: Pteromalidae) preferred *S. zeamais* than *Rhyzopertha dominica* to feed on. This resulted in confirming that *A. calandrae* is good for implementing the use of this natural enemy as a control tool. It was also found that the entomopathogenic fungus *Beauveria bassiana* can be used as a biocontrol agent against the maize weevil [88]. Despite the benefits of biological control, though, the methods have taken some time to catch on globally.

## 6. Pesticide Resistance

The most effective method of controlling an attack by the maize weevil and the ensuing damage in developing countries is currently the use of conventional synthetic pesticides [89,90]. Synthetic pesticides may also play a significant role in reducing storage losses due to insect pests [91,92]. In this 21st century, global food security threatens human survival, and the development of pesticide resistance is contributing to this issue. Indeed, pesticide resistance development is becoming a bigger issue for the protection of stored goods [63], and global reports of stored goods and insect pesticide and resistance are actually becoming prevalent [93,94]. Pest management costs rise as a result of increased pesticide use, which also increases risks to human health and environmental damage. Finding safer substitute control methods, such as the use of biological agents against stored grain pests, is thus important [63]. By 1996, there were more than 600 kinds of plant-eating insect pests that were resistant to insecticides [95]. A growing body of evidence has reported resistant strain of pests in staple foods, such as cowpea and maize [96,97], resulting in secondary pest outbreaks [98]. Another study has reported that most storage buildings are vulnerable to pest re-infection and require shorter intervals between pesticide applications due to pest re-infection, and poor storage facilities in underdeveloped nations make the use of synthetic chemicals ineffective and contribute to insect resistance to chemicals [99]. To better comprehend this, consider that insect populations become resistant to pesticides when the same chemical is applied repeatedly over the course of the pest’s life cycle [100]. Additionally, when a pesticide develops resistance, it is typically followed by the use of new products, larger dosages of the original pesticide, or an increase in pesticide treatments.

Studies on pesticide resistance are crucial for developing practical pest management strategies as well as serving as models for the evolution of newly adaptive phenotypes and the physiological (and genetic) alterations that they cause [101,102,103]. When pesticide use is reduced, eliminated, or changed, the pleiotropic consequences that typically follow selection for pesticide resistance may disadvantage the resistant insects [101,104,105,106,107]. In the absence, diminution, or modification of pesticide selection, this may cause a gradual decrease in the frequency of resistant individuals over time. Only in the last scenario is there no common expectation of a physiological cost of resistance among the four fundamental molecular mechanisms of pesticide resistance (constitutive overproduction, constitutive underproduction, target alteration, and inducible change in gene regulation) [98,99,101,106]. Pesticide resistance-causing changes typically cause negative effects by obstructing the routes of their targets (or receptors) [99].

## 7. Medicinal Plants as a Natural Approach to Technical Control of Maize Weevil

Currently, sustainability and feeding a growing global population has spiked an overwhelming concern worldwide. Research on alternative eco-friendly and sustainable methods to control pests has become necessary due to public concerns about the toxicity of pesticides and their effects on both the environment and on public health [62,108]. Many studies agree that there is a need to develop alternative strategies with low adverse effects on consumers and less persistent effects on the environment [109,110,111]. The search for secure, efficient, and practical substitutes has been more intense due to worries about the effects of synthetic pesticides on human health and on the environment. In this respect, plant-based pesticides may be less harmful, quickly biodegradable, appropriate for use by small-scale farmers, and capable of preventing pests from damaging grain [112]. More than 2500 plant species from 235 families have been documented in prior research, and they have proven to have biological activity against various pests [113]. Despite this, few have been marketed for commercial usage or referred to as farming products [114]. Numerous herbal remedies, including plants, essential oils, and their chemical components have been documented for their inhibitory effects on insect pests [115,116]. Non-target organisms are at a lower risk because of plant-derived pesticides due to the fact that they are short-lived in the environment [117,118]. Due to the fact that they occur naturally, certain consumer groups and organic certification programs have easily warmed up to them.

Different plant-based treatments are used by smallholder farmers to manage pests in storage grains. The specific approach taken by these farmers varies from location to location and appears to be influenced by the kind and potency of useful resources that are available in various places [119]. For instance, *Lippia javanica*, *Eucalyptus* spp. and *Tagetes minuta* is utilized by farmers as a botanical pest control [120,121], and it can be emphasized that many African plants are an excellent source of pesticide. They have been shown to contain either antifeedant, repellents, or insecticidal compounds produced by plants in order to protect stored products. Another study showed that African plants with a repelling effect have not yet gained much publicity [122]. Many efforts have been made to screen plants with better botanical pesticides, which can be used as an alternative to synthetic pesticides [123]. Indeed, African small-scale farmers have been controlling insect infestation for a long-time using plants [124,125,126]. Due the overpriced tag associated with synthetic pesticides, struggling counties such as Zimbabwe cannot easily access or afford them, and even though relatively little is known about their effectiveness, such farmers have employed local herbs and other therapies in order to combat storage pests [127]. The majority of flora are accessible to farmers nearby, and they are environmentally beneficial because they have no aftereffects. Plants are quick to interrupt the eating of an insect pest, frequently resulting in an immediate paralysis or cessation of feeding, even if the insect may not die for several hours or days [55].

According to Tapondjou et al. [75], mixing stored grains with leaf, bark, seed powder, or essential oil extracts of plants reduces oviposition rate, suppresses progeny production, and is toxic to adults, which ultimately results in low infestation and yield losses. At economically sound doses, grain protectants should be investigated. Moreover, the propensity to test the substance at high-application concentrations is another common flaw in screening plant materials for insecticidal efficacy [36]. It was noted that a plant powder concentration used to protect grain against the pests found in stored products should not surpass 2.0% weight/weight in order to be economically justifiable. Making sure that only the bare minimal amount of plant material is used in order to produce the desired result is essential to preventing harmful impacts of plant materials on farmers, consumers, and the environment [128]. Most farmers are ignorant of the right way of using synthetic chemical pesticides, the toxicity effects as well as the best concentration. Therefore, plant-derived pesticides are a great alternative. [60]. This has prompted pest control research on how to properly use plant-based insecticides as well as concentration control.

For example, Ivbijaro et al. [125] reported that the toxicity of *Piper guineense*, a black pepper variety, is very high on the cowpea beetle, *Callosobruchus maculatus.* Furthermore, Ivbijaro et al. [126] revealed the use of neem seed powder, *Azadirachta indica*, on weevil-infested maize grains eliminated oviposition at the high dose, significantly reduced oviposition at the medium dose, and totally stopped post-embryonic growth at all doses. Because of this, all doses of groundnut oil greatly increased the mortality of the rice weevil *Sitophilus oryzae* while decreasing its natality and oviposition [127]. Citrullus vulgaris testa powder has been shown to drastically decrease *S. zeamais* natality and oviposition in maize [15]. *Callosobruchus maculatus, C. chinensis,* and *C. rhodesianus* oviposited much less frequently, and the lifespan of adults of both *C. maculatus* and *C. chinensis* was greatly reduced by corn oil, groundnut oil, sunflower oil, and sesame oil [128]. High adult weevil mortality for *S. zeamais* has been reported with leaves from *Eucalyptus globules, Schinese molle, Datura stramo-nium, Phytolacca dodecandra,* and *Lycopersicum esculentum* as an effective treatment [129]. Drawing from the above, it can be suggested that pest infestation of grains in storage can be reduced by using natural plant products plants, providing golden opportunities to explore and develop natural plant-based pesticides utilized to prevent pest infestation of storage goods [63]. Plant powders, according to Ukeh et al. [130], are inexpensive, readily available, and easily biodegradable. They also will not contaminate food goods by serving as protectants in small-scale storage systems. In addition, medicinal plants are attacked by insects and mites. Table 1 demonstrates various medicinal plants, their preparation, and their biocompounds.

The toxic, lethal, repellent, antifeedant, fumigant, growth-regulating, and deterrent effects of medicinal plants and their derivatives or extracts have been assessed for various pest control strategies. Furthermore, botanical insecticides can influence both behavioural and physiological processes in contrast to conventional insecticides, which are based on a single active ingredient [144]. *Acanthus montanus, Acanthospermum hispidum, Argyreia nervosa,* and *Alchornea laxiflora* powders successfully prevented the ability of the maize weevil, *S. zeamais*, to lay eggs in a study by Ileke [132] in 2020. *S. zeamais* lay considerably fewer eggs on treated maize grains than on untreated seeds. According to studies, complex combinations found in plant powders, extracts, or oils suppress the activity of the acetyl cholinesterase enzyme (AChE) [145]. Acetyl cholinesterase enzyme (AChE) activity is inhibited, which can interfere with the neuromodulator octopamine and kill pest insects by blocking their GABA-gated chloride channels [146]. Contact toxicity may also have contributed to the high percentage of *S. zeamai* deaths after exposure to the studied plants’ leaf powders and extracts. Most insects breathe using a trachea, which typically opens through spiracles at the body’s surface. According to Adedire et al. [81], these apertures or air chambers may have been prevented from allowing adequate oxygen into the bodies of the insects, which ultimately caused their asphyxiation and death.

To increase the toxicity of insecticides against insect pests, biochemical and physiological characteristics could be evaluated in addition to the lethal effect. The activity of acetycholinesterase (AChE) and the vetillogenin body protein are just two examples of these metrics [147]. When the synthesis of oocytes during vetillogenesis fails due to a lack of protein, it has been discovered that the ability of many insects to reproduce is affected [17,148]. Acetylcholine is broken down by AChE at the synaptic cleft in order for the subsequent nerve impulse to travel across the synaptic gap, acting as a neurotransmitter [149]. When this action is disrupted, a loss of coordination is likely to happen, which could cause the insect to be knocked down and subsequently die. According to Abdullahi et al. [150], the phytochemical components of plant powder may kill insects through a variety of mechanisms, including contact poisoning, interference with acetylcholine receptors [151], and ingestion of the powder’s constituents, which may then disrupt the insects’ metabolic processes and result in a quick demise [152].

## 8. Conclusions

To feed a growing global population, crop protection is one of the keys to ensuring food security. Specifically, in the agricultural sector, the maize weevil causes qualitative and quantitative damage to stored grains, cereals, and other processed and unprocessed stored products. Global concern exists over the use of pesticides to control stored goods, though, due to concerns over environmental contamination, human health, insect resistance, and chemical residues, as well as the high costs. There is now a consensus that new, affordable, easily biodegradable, eco-friendly, and sustainable pesticides are needed in order to control the maize weevil. Plant-based pesticides are an excellent candidate since they possess an extensive range of biological activities against insect pests. In this context, medicinal plants fulfil the requirements toward a sustainable greener agriculture, thereby ensuring that we alleviate the food crisis and solidify global food security.

## Figures and Tables

**Table 1 plants-12-02505-t001:** Medicinal plant used as pest control of *Sitophilus zeamais*.

Medicinal Plant	Biocompounds	Preparation Plant	Ref.
*Annona muricata*	Alkaloids, flavonoids, tannins, and saponins	Methanol extract	[131]
*Acanthus montanus*	Alkaloids, saponin, tannin and flavonoid	Plant powder	[132]
*Zingiber officinale*	N/A	Plant powder	[133]
*Piper corcovadensis*	Phenylpropanoid, monoterpenes α-pinene, and terpinolene	Leaf essential oil	[134]
*Alchornea cordifolia*	N/A	Plant powder	[135]
*Lantana camara*	Phytol, Pyrroline, Paromomycin, Pyrrolizin, and 1-Eicosano	Plant powder and essential oil	[136]
*Lamium purpureum*	Alkaloids, terpenoids, flavonoids, tannins, saponnins, phytosteroids, and phenolic compounds	Plant powder	[137]
*Cupressus macrocarpa*	N/A	Plant powder	[138]
*Moringa oleifera*	Alkaloids, saponins, tannins and phenolic, steroids, flavonoids, anthraquinones, phlobatannins, cardiac glycosides, and terpenoids	Plant powder	[139]
*Eucalyptus camaldulensis*	N/A	Plant powder	[140]
*Tithonia diversifolia*	Tannin, flavonoid, saponin, phenol, terpenoid, glucosides, and alkaloid	N-hexane extracts	[141]
*Eucalyptus citriodora*	Saponins, tannins, flavonoids, phenols, quinones, and alkaloids	Essential oil	[142]
*Foeniculum vulgare*	Anethole, fenchone, d-limonene, alpha-pinene, and p-cymene	Essential oil	[143]

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
