# Peer review of "Medicinal Plants as a Natural Greener Biocontrol Approach to “The Grain Destructor” Maize Weevil (Sitophilus zeamais) Motschulsky"

_plants, 2023, doi:10.3390/plants12132505_

Round 1

Reviewer 1 Report

The maize weevil (Sitophilus zeamais), is a primary field and store pest of maize, producing huge post-harvest losses worldwide. Different synthetic insecticides are currently used to control maize weevil. In this context, medicinal plants have been proposed as environmentally friendly alternatives to conventional insecticides. 

The present review lacks in-depth elaboration of the effect mechanism of medicinal plants on S. zeamais. Even though medicinal plants may exert their individual effect against the target insects, the bioactivity of medicinal plants usually depends on their components since synergistic, antagonistic, or additive interactions can occur among them, which ultimately influence their insecticidal properties. In addition, the insecticidal effect of pesticides depends on the method of application used against the insect.  Plant-derived pesticides enter the insect body through the respiratory system? Previous some studies evaluated the effect of phenylpropanoids on S. zeamais that their toxicity may be due to ingestion and digestion of the compounds in the stomach and not by contact and absorption through the cuticle.

Minor editing of English language required

Reviewer 2 Report

Very interesting study. Plant-based pesticides are excellent alternatives to chemical pesticides against insect pests.

Reviewer 3 Report

The manuscript has multiple issues:

-First of all, it doesn't focuses on the topic mentioned in the title 

-The style of it doesn't conform with scientific standards (e.g. "skyrocketing population)

-Different topics are mixed up and repeatedly recurring in the whole text: they have to be arranged in a more logical order

-The process how the sources were collected and evaluated is not clearly presented

-Excessive discussion of topics unrelated with the title of the manuscript (e.g. l 202-228, 249 - 277, and many others)

-There are many factual mistakes or misinterpretations in the text

Some (but far not all) specific examples:

l. 3-4: (Sitophilus zeamais ((Mochul'skii))

l.31. "two insects per grain generate 18.3% losses in 48 days. [2]." 2 insects / grain produce would produce total loss; the cited reference is about essential oils

l.34: Aspergillus flavus Link.

l.39 - 40: Unfinished sentence

Figure 1: looks like an explonation from a student textbook. Some statements of it are incorrect or can be questioned

l.103, and later: auktor name and taxonomy have to be written at the first mentioning of the scientific name.

Table 1. is not "Host plant distribution" but some arbitrary examples of host plants

Table 2. seems arbitrary, it misses any logical structure

l. 139 - 141: an arbitrary list of fungal species and genera: many of them are not plant pathogens

l.144: "green fungus": it is a medical term for human and animal Aspergillosis

The language of the text needs a major revision.

Author Response

No changes were suggested, thus no changes were made. 

Round 2

Reviewer 1 Report

The Authors have improved the manuscript according to the comments. It can be accepted in current form.

Author Response

No corrections.

Reviewer 3 Report

The English of the manuscript has been greatly improved, although minor errors can still be found (eg. line 502: instead of "death effect", "lethal effect")

Some objection has not been considered: Table 1. is pointless with misleading informations; Figure 1. is misleading with false and overly simplified; both should be deleted.
